# Integrating Smoking Cessation Care into a Medically Supervised Injecting Facility Using an Organizational Change Intervention: A Qualitative Study of Staff and Client Views

**DOI:** 10.3390/ijerph16112050

**Published:** 2019-06-10

**Authors:** Eliza Skelton, Flora Tzelepis, Anthony Shakeshaft, Ashleigh Guillaumier, William Wood, Marianne Jauncey, Allison M. Salmon, Sam McCrabb, Billie Bonevski

**Affiliations:** 1School of Medicine and Public Health, The University of Newcastle, Faculty of Health and Medicine, 1 University Drive, Callaghan, NSW 2308, Australia; Flora.Tzelepis@hnehealth.nsw.gov.au (F.T.); Ashleigh.Guillaumier@newcastle.edu.au (A.G.); Sam.Mccrabb@newcastle.edu.au (S.M.); Billie.Bonevski@newcastle.edu.au (B.B.); 2Hunter New England Local Health District, Hunter New England Population Health, Booth Building, Longworth Avenue, Wallsend, NSW 2287, Australia; 3National Drug and Alcohol Research Centre, The University of New South Wales, 22-32 King Street, Randwick, NSW 2031, Australia; a.shakeshaft@unsw.edu.au; 4Uniting, Sydney Medically Supervised Injecting Centre, 66 Darlinghurst Road, Kings Cross, NSW 2011, Australia; wwood@uniting.org (W.W.); mjauncey@uniting.org (M.J.); allisonsalmon@gmail.com (A.M.S.)

**Keywords:** organizational change, tobacco smoking, facilitators, barriers, persons who inject drugs, supervised injecting facility, smoking cessation, acceptability

## Abstract

*Background:* Clients accessing supervised injecting facilities (SIFs) smoke at high rates. An SIF piloted an organizational change intervention to integrate smoking cessation care as routine treatment. This study aims to explore staff acceptability, perceived facilitators, and perceived barriers to implementing six core components of an organizational change intervention to integrate smoking cessation care in an SIF. Staff and client views on the acceptability, facilitators, and barriers to the provision of smoking cessation care were also examined. *Methods:* This paper presents findings from the qualitative component conducted post-intervention implementation. Face-to-face semi-structured staff interviews (*n* = 14) and two client focus groups (*n* = 5 and *n* = 4) were conducted between September and October 2016. Recruitment continued until data saturation was reached. Thematic analysis was employed to synthesise and combine respondent views and identify key themes. *Results:* Staff viewed the organizational change intervention as acceptable. Commitment from leadership, a designated champion, access to resources, and the congruence between the change and the facility’s ethos were important facilitators of organizational change. Less engaged staff was the sole barrier to the intervention. Smoking cessation care was deemed suitable. Key facilitators of smoking cessation care included: Written protocols, ongoing training, and visually engaging information. Key barriers of smoking cessation care included: Lack of access to nicotine replacement therapy (NRT) outside of business hours, practical limitations of the database, and concerns about sustainability of NRT. *Conclusion:* This study develops our understanding of factors influencing the implementation of an organisational change intervention to promote sustainable provision of smoking cessation care in the SIF setting.

## 1. Introduction

Approximately 12 million people worldwide inject drugs [1]. Among people who inject drugs, tobacco smoking is one of the most commonly used substances with prevalence estimates ranging from 84% [2] to 90% [3]. Despite this, people who inject drugs consistently report high levels of interest in quitting smoking [3].

Supervised injecting facilities (SIFs) are accessed by a large number of people who inject drugs and provide basic treatment in addition to facilitating access to other health care services [4]. Such services may also be able to integrate smoking cessation care into routine treatment though the acceptability of such care. Organisational change is proposed to facilitate the provision of smoking cessation care into routine delivery in healthcare services [5]. Organisational change interventions, such as the Addressing Tobacco Through Organisational Change [6,7] and Systems Change Approach [8], have been developed for increasing smoking cessation care delivery in the alcohol and other drug (AOD) setting.

To address the limited knowledge of an organisational change intervention and the provision of smoking cessation care in the SIF setting, we conducted a pre- and post-test pilot study [9]. An organisational change intervention was implemented. The intervention consisted of six core components: Engagement of organisational support, identification of a support champion, provision of education and resources, provision of evidence based treatments, creation of a system to assess and record smoking status, and promotion of centre policies that support and provide tobacco dependence services. From pre- to post-intervention, staff reported smoking cessation care practices and client reported receipt significantly increased [9]. Whether the organisational change intervention was an acceptable means to ensuring the treatment of tobacco smoking in the SIF for staff is unknown as are facilitators and the challenges faced. Further, the acceptability, facilitators, and barriers to smoking cessation care for staff and clients in the SIF setting are unknown.

Prior quantitative research has examined the acceptability, facilitators, and barriers to smoking cessation care from staff and client perspectives in the broader AOD setting [10,11,12,13,14,15,16,17]. Factors have been identified that occur at the service [11,12,14], staff [11,13,16], and client [15] levels. To date, only one qualitative study by Pagano and colleagues [17] has identified a number of facilitators (e.g., enhanced leadership, funding, and smoke-free policy) and barriers (e.g., staff smoking, environmental barriers, smoking culture, client resistance, and lack of resources) to smoking cessation care. Given the lack of research evidence, the present study aims to explore the acceptability, facilitators, and barriers as viewed by SIF staff regarding the organisational change intervention to integrate smoking cessation care. Staff and client views on the acceptability, facilitators, and barriers of smoking cessation care at an SIF were also explored.

## 2. Methods

### 2.1. Design

This paper reports on the qualitative component of an organisational change intervention to integrate smoking cessation care into routine delivery at an SIF [9]. The intervention is described elsewhere [9]. The current study was conducted from September to October 2016 one year after the post-intervention phase and is based on semi-structured manager and staff interviews as well as focus group discussions with clients. This study was reviewed and approved by the University of Newcastle Human Research Ethics Committee (Approval Number H-2013-0082).

### 2.2. Sample

Eligible managers and staff were current employees who had worked at least one shift every two weeks at the facility during the study period, and had therapeutic client contact in a role where treatment was part of their normal duties. Eligible clients were current clients aged 18 years or over; able to read, speak, and comprehend English; had at least one visit to the facility in the last 12 months (during the intervention period); and self-identified as either a current tobacco smoker or recent quitter (in the last 12 months).

### 2.3. Procedures

The site contact disseminated an email invitation with a participant information statement attached to all members of staff (variety of management and staff roles, those who work weekdays and in the weekend for a representative sample). Individuals interested in participating were able to email or call to register interest in participating in the interviews. From the 16 eligible managers/staff invited, 14 individuals (3 managers and 11 staff, response rate 87.5%) participated in the interviews conducted by one member of the research team (ES). The interview guides (see Appendix A) engaged participants in discussion about each of the six components of the organisational change intervention to integrate smoking cessation care into routine delivery, and the acceptability, facilitators, and barriers to addressing client tobacco smoking. Participants also completed a brief survey about demographics and smoking history prior to the commencement of the interview.

A flyer promoting the client focus group was displayed in the waiting room and after care rooms of the facility. Management and staff were involved in the promotion of the focus groups with clients, providing participant information statements to the client on entry to the service. From the 11 eligible clients invited, 9 participated (response rate 82%). Two client focus groups (*n* = 5 and *n* = 4, respectively) were held in a private meeting room at the SIF. Group discussions were facilitated by two female members of the research team employed by the University of Newcastle, a post-doctoral research academic (AG) and a doctoral candidate (ES), both of which had prior experience in working with vulnerable groups. Each focus group allowed a semi-structured format to engage participants in discussion about their smoking, experiences with receiving smoking cessation care, and the acceptability of staff providing such care whilst individuals were utilizing the service (see Appendix A). Participants also completed a brief survey prior to the commencement of each focus group. Individuals participating in client focus groups were reimbursed for their time with a $10AUD supermarket gift card. Written consent was gained for the demographics survey and audio recording prior to study commencement.

### 2.4. Data Analysis

All interviews and focus groups were audio-recorded and transcribed verbatim. Staff and client transcripts were merged and imported into qualitative data analysis software, NVivo version 11, for analysis. The thematic analysis was based on procedures described by Braun and Clarke [18]. This process involved an initial reading of the data for accuracy and overview, followed by the development and application of descriptive codes based on patterns observed in the data and critical analysis of codes to collate them into themes. Initial coding and data analysis were conducted by ES. A second author (SM) coded and analysed one-third of all transcripts. ES and SM met to discuss emerging themes. After each transcript was coded individually, content from both staff and clients was merged into higher order themes and sub-themes based on acceptability, facilitators, and barriers. Disagreement in coding or data categorization was revolved by discussion. Participant characteristics are described using frequencies and percentages for categorical data and means and standard deviations for continuous data using STATA version 13 (StataCorp LP, College Station, TX, USA).

## 3. Results

Manager interviews ranged from 51 to 120 min while staff interviews ranged from 25 to 39 min. Manager and staff characteristics are provided in Table 1. Nine clients took part across two focus groups. Discussions ranged from 33 to 51 min. Client characteristics are outlined in Table 2.

### 3.1. Acceptability of Six Core Components of an Organisational Change Intervention

This theme explores specific intervention components from manager and staff perspectives, including the suitability and adaptations made to the facility.

Creation of a smoker identification system. Managers and staff felt the changes to the electronic database to identify current tobacco users were positive as the change was viewed to have a limited impact on current duties. In particular, staff thought that the single question asking smoking status was acceptable and feasible, particularly in terms of the brief nature of asking whether the person currently smokes tobacco:
“It’s almost like you could do it as a throw away question, you just ask it as part of your stuff”(Staff member 5)

Further, staff thought that the timing was appropriate (Stage 1) when other substance use was assessed. This sentiment was further shared when staff reflected on the mandatory nature of the assessment question, and that it triggers a useful reminder prompt which they viewed to be of assistance:
“It’s good… because it’s a good prompt”(Manager 2)

Providing evidence based tobacco treatments. Importance was placed on the suitability and feasibility of NRT to be delivered to clients:
“It was pretty seamless in the sense of it just slotting in to what we normally do”.(Staff member 8)

The provision of this care became part of usual processes and staff considered these duties as their role despite recognizing that there had been no formal change to their position description:
“Well, it’s just part of our, well not our job description, but it’s part of our job in Stage 3”(Staff member 11)

Providing education and resources. The training engaged staff in the organisational change process and the content was found to be appropriate:
“That training day got people really engaged in the project I think. It was a really good way of introducing it and getting people on board.”(Manager 2)

Identification of a support champion. Participants agreed that designating one individual at the facility as the support champion was suitable. Many thought that the support champion should be a continued role at the facility and could be integrated into the position description of a staff member.
“I know the person to go to would be [the support champion]. Like he would be able to answer any question I had.”(Staff 1)

Engagement of organisational support. Staff perceptions of the intervention was that it was deemed acceptable, and congruent with the current goals and mission of the facility’s harm minimization ethos as nicotine dependence is a drug addiction. Overall, when reflecting on the intervention, both managers and staff felt that it became part of the facility to provide this care:
“You wouldn’t even call it a research project anymore, you know, it’s just what we do. It’s just part of our daily work.”(Staff member 7)

Promote smoke-free policy. Managers and staff noted that the facility had a total ban smoking policy, however, direct changes to this policy during the intervention period were not reflected upon. All individuals viewed that ensuring compliance to the smoking policy was part of their job and noted that they relied on security staff for enforcement of the policy.

### 3.2. Facilitators of Organisational Change

This theme placed value on dedicated individuals, commitment from leadership, and access to resources.

A dedicated individual. Managers and staff expressed that the support champion was instrumental in ensuring the intervention remained an organisational priority and was smoothly incorporated into daily duties. The success of the intervention was aligned with beliefs of a support champion, who was committed and contributed to building the self-efficacy of other staff. A dedicated individual or group of individuals was regarded as key to embracing the intervention. Staff also viewed that it was important to have someone at the start of the intervention when things were commencing to keep the project at the forefront of people’s mind.
“I suppose we could have achieved it, but I think we’ve achieve it better having [a support champion] in that role.”(Staff member 1)

Training and support. Managers and staff expressed that the delivery of information fostered supportive attitudes about the intervention. A small subset of staff described their feelings about the intervention as “*not overly committed”* and “*sceptical”*. Despite articulating negative attitudes at the start of the intervention, many felt that their views changed once they received smoking cessation training. Staff began to view the intervention as *“worthwhile”*.

Access to resources. Managers and staff agreed that having access to multiple types of NRT removed financial barriers both for the facility and the client, and facilitated greater engagement in the intervention. Many staff thought that without access to free NRT it would have been harder to motivate staff to provide care.

Leadership commitment. There was a strong sense of commitment from managers and willingness for the intervention to continue beyond the research phase:
“We’re not just going to pilot; we’re going to keep with this.”(Manager 1)

Management commitment to the intervention resonated in staff interviews with many viewing that the intervention was backed completely by the decision makers at the facility. One staff member noted:
“It was going to become our mainstream practice.”(Staff member 6)

### 3.3. Barriers of Organisational Change

This theme emphasized the disconnect between staff who are less frequently at the SIF and the change.

Less engaged staff members. A consistent concern among managers was that some staff were not motivated to embrace the changes. Managers identified these individuals as those who were less frequently at the facility, in casual or part-time positions. Staff also affirmed the barriers faced by managers, with some proposing that this was due to working mainly on the weekend when there was less support. Other staff felt as though they became less engaged when moving from full-time to part-time, making the changes more difficult to sustain:
“It was easy at the beginning [when I was full-time], but then it became hard… you lose contact with what’s happening.”(Staff member 10)

### 3.4. Acceptability of Smoking Cessation Care

This theme found that the treatment of tobacco smoking at this facility was suitable and appropriate.

Finding the right time. Managers and staff agreed that providing multiple types of NRT was acceptable as it promoted choice as well as opportunity to their clients. The *“right time”* was viewed as stage 3. Staff referred to this area as a “*safe place*” where clients would be most receptive to addressing their tobacco smoking as they were most open to talking *“about their life and about change in their future”*. Acceptance of smoking cessation care as part of the facility’s role was consistently mentioned with staff members noting:
“We’ve been doing it for a while now, so its (providing smoking care) kind of embedded in a way in what we’re doing.”(Staff member 9)

Clients agreed that the provision of NRT was suitable and should be part of routine care. Clients felt well supported and well informed to be using the different types of NRT:
“They’ll actually talk to you about it and you can sit down and they’ll take the time to explain everything.”(Client 7)

While clients suggested that care was mostly initiated by staff, some clients recalled that they became more proactive about asking staff for NRT and that they felt comfortable doing so.

Clients were happy for the treatment of tobacco smoking to be a facility-wide approach, acknowledging that it was appropriate for all staff to provide this care. Clients linked the appropriateness of receiving this care at the facility as their only contact with medically trained staff:
“For a lot of us this is the only place that we come into contact with any nurses or any health care workers.”(Client 4)

### 3.5. Facilitators of Smoking Cessation Care

This theme highlighted the importance of guidelines and protocols for staff to support the provision of care, ongoing staff training, and visually engaging campaigns for clients.

Supporting staff to support the clients. Staff asserted that their provision of smoking cessation care to clients was supported by written protocols that ensured consistency and continuity. Staff felt that these guides also assisted them in feeling comfortable to treat clients.
“We sort of have a bit of a package of how to talk to people about stopping smoking which I find really useful.”(Staff member 4)

Clients understood that there were set protocols in place by the facility and its staff in terms of the amount of NRT that was available. Clients deemed the provision of NRT was adequate and that the process in which it was provided could not be faulted:
“You’re given a set amount which is quite adequate and you just come back the next day to get a refill.”(Client 5)

Staff noted that ongoing education was always readily available, that they felt well supported, and that it assisted them to continue to provide smoking cessation care to clients. Paper-based resources, including information about the various types of NRT, were regarded as useful in facilitating care provision for clients. Staff noted that the ongoing training was predominately provided by the support champion:
“To go ‘oh I’m not too sure how to ask that’…just to reflect on it, have a chat with [the support champion] it’s been good.”(Staff member 9)

Visually engaging campaigns are important. Staff noted that refurbishing the stage 3 area with striking posters that included information and graphics about the health consequences of tobacco smoking as well as a “no smoking doll” facilitated the conversation about treatment of tobacco smoking and subsequent provision of smoking cessation care:
“Quitting smoking themed dolls… that caught a lot of clients’ eyes, and so then it would start the conversation about NRT.”(Staff member 6)

Clients recalled that there were coloured posters in the injecting booths and also in the stage 3 area. Throughout discussions, clients commented on their impaired memory and the importance for reminders either from staff or visual aids were good and ensured that they received care:
“It reminds you. Because if you’re using you forget a lot.”(Client 8)

### 3.6. Barriers to Smoking Cessation Care

This theme emphasized a share frustration by staff and clients at the lack of smoking cessation care, such as NRT, outside of normal business hours, staff concerns regarding practical limitations of the current smoker identification system, and concerns about the sustainability of resources.

Lack of smoking cessation care resources. Even though staff saw the provision of smoking cessation care as part of their role and not just solely the responsibility of one person, they also viewed that one person needed to be the co-coordinator. They viewed the importance of having someone ensure there was sufficient stock available for clients. The lack of NRT out of normal business hours was a major barrier for both staff and clients:
“If it’s on the weekend we’ve got no access to any of the resources, they’re all locked away.”(Staff member 7)
“I wanted to get inhalers, and it may have been on a weekend and I couldn’t.”(Client 9)

Practical limitations of the smoker identification system. Staff viewed failures in the provision of care to be aligned with limitations of the database. Given that at stage 1 smoking status was recorded as no if someone was making a quit attempt and had stopped smoking, the system did not provide prompts for this situation and so the opportunity for relapse prevention care was missed:
“We couldn’t record that, because they had not said they were tobacco smokers in Stage 1.”(Staff member 9)

Concerns about sustainability. Managers and staff associated their initial apprehensions about providing smoking cessation care with concerns of longevity. Although tobacco smoking was seen as something that should be treated, there were sustainability concerns:
“I was thinking about a project running like this, which introduces clients to NRT, and then disappears. And then clients are left to their own devices.”(Staff member 3)

Clients’ echoed concerns regarding funding, noting that they would not be able to afford to buy NRT nor access subsidized NRT given that many were homeless and without a Medicare card were unable to receive treatment elsewhere.

## 4. Discussion

This is the first qualitative study to explore the implementation of six components of an organisational change intervention to integrate smoking cessation care into routine delivery at an SIF. Our results suggest that the intervention components were acceptable to staff members within this setting and that treating tobacco smoking was appropriate for both SIF staff and their clients.

Consistent with an existing qualitative study of organisational change in healthcare settings [19], which shows the importance of an influential and current employee, staff reported that a support champion was essential to ensuring the successful maintenance of the intervention. Effective support champions are not those who just implement the intervention but serve as a local expert [20]. Staff consistently reported that the support champion was important for further questions and practical considerations.

Similar to Pagano and colleagues [17], support to provide NRT was a facilitator and lack of resources was a barrier to smoking cessation care. Possessing NRT was seen as a necessary and generally sufficient means to ensure both staff and clients addressed tobacco smoking during engagement with the facility. Our results suggest that care was supported by policy-level changes with the implementation of written guidelines and protocols for care. Comparative to prior research [17], our study did not identify client resistance or staff smoking as barriers to smoking cessation care, highlighting that the attitudes and treatment culture appear to be supportive of the treatment of tobacco smoking.

This study also found a number of novel findings. Both staff and clients expressed that the use of visually engaging campaigns facilitated the provision of care. Visually engaging campaigns, such as posters and dolls, with NRT information and the harms of tobacco smoking facilitated smoking cessation care provision. Ensuring information is striking to both staff and client is important and therefore further exploration of the effectiveness of such resources may be necessary.

### 4.1. Implications

This study suggests an organisational change intervention can be used in settings that are not delivering smoking cessation care or as a way to increase delivery for those with sub-optimal provision. The components of the intervention that were viewed consistently as acceptable were the identification of a support champion and the provision of evidence based treatment, such as NRT. This may be a good place to commence an organisational change intervention to increase the provision of smoking cessation care. Changes to an existing system, such as the electronic health record, to ensure smoking status is screened and smoking cessation care is recorded can be individualized and implemented in a way that works with the existing systems and workflows of the facility wanting to make changes.

Service directors contemplating the implementation of an organisational change intervention should be aware of the barriers that they need to account for when choosing this type of intervention. Drawing upon the healthcare reform literature [21], suggested strategies to address barriers, such as a lack of employee engagement, involve specific drivers: Staff are involved in the decision-making, the facility provides encouragement for development, and two-way dialogue with facility leaders and co-workers to create a reliable support team [22]. Providing these opportunities in an effort to address barriers, such as lack of engagement, will allow for services to efficiently implement the new processes and procedures within their setting [19].

### 4.2. Study Strengths and Limitations

It is important to consider the results within the context of the study’s strengths and limitations. First, the participants were all staff and clients from a single Australian SIF, thus their experiences may not occur within other facilities located nationally or internationally. Second, participant responses may have been influenced by social desirability bias. Participants may have been reluctant to openly criticize the intervention or treatment of the client’s tobacco smoking. However, social desirability bias appears to have been minimal given the willingness of staff and clients to discuss their personal experiences, both positive and negative, with the intervention.

## 5. Conclusions

SIFs appear to be appropriate settings for the provision of smoking cessation care and therefore may have success in reducing tobacco related morbidity and mortality experienced by people who inject drugs. Future research may be interested in evaluating the effects of an organisational change intervention on both behavioural outcomes (quit rates) and process outcomes (provision of smoking cessation care).

## Figures and Tables

**Table 1 ijerph-16-02050-t001:** Manager (*n* = 3) and staff (*n* = 11) demographics and smoking characteristics.

Characteristic	*n*	%
Age (average, SD)	46.3 (11.3)	
Gender		
Female	9	64.3
Male	5	35.7
Role		
Manager	3	21.4
Health Education Officer	6	42.9
Nurse (registered, endorsed enrolled)	5	35.7
Employment status		
Full-time	4	28.6
Part-time	10	71.4
Number of years in the alcohol and other drug field (average, SD)	16.64 (8.9)	
Number of years at Medically Supervised Injecting Facility (average, SD)	7.5 (3.5)	
Smoking status		
Daily	3	21.4
Occasional	1	7.1
Ex-smoker	6	42.9
Never-smoker	4	28.6
Quit intentions		
Quit in the next 30 days	1	25
Quit, but not in the next 6 months	1	25
Don’t know	2	50

**Table 2 ijerph-16-02050-t002:** Client demographics and smoking characteristics (*n* = 9).

Characteristic	*n*	%
Age (average, SD)	50.2 (7.4)	
Gender		
Female	2	22.2
Male	7	77.8
Indigenous status		
Aboriginal	2	22.2
No	7	77.8
Marital status		
Married	1	11.1
De facto or living with partner	1	11.1
Separated or divorced	1	11.1
Never married or single	4	44.5
Widowed	2	22.2
Education		
Primary school/ High school	6	66.7
Trade or trade qualification	1	11.1
University	2	22.2
Income (personal, per week)		
Between $100–$199	1	11.1
Between $200–$299	2	22.2
Between $300–$399	2	22.2
Between $400–$499	1	11.1
More than $500 per week	1	11.1
Prefer not to answer	2	22.2
Smoking status		
Daily	7	77.7
Occasional	2	22.2
Heaviness of smoking index		
Low	3	33.3
Moderate	4	44.5
Heavy	2	22.2
Quit intentions		
Quit in the next 30 days	3	33.3
Quit in the next 6 months	2	22.2
Quit, but not in the next 6 months	2	22.2
Don’t know	2	22.2

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
