# Peer review of "Integrating Smoking Cessation Care into a Medically Supervised Injecting Facility Using an Organizational Change Intervention: A Qualitative Study of Staff and Client Views"

_ijerph, 2019, doi:10.3390/ijerph16112050_

Reviewer 1 Report

The concurrent use of illicit substance and tobacco is an important, yet under-researched public health issue.  Because of this, the present paper is of relevance and is likely to be a topic of interest to readers of the journal.

The paper describes a qualitative exploration staff and client perceived barriers and facilitators, and acceptability perceptions of a smoking cessation intervention in a supervised injecting facility.  Results are obtained from interview and focus groups. Overall, I found the paper to be generally well written and I am confident that the study design is appropriate for the research purpose. 

Minor revisions

1.        Please spell out abbreviations in the first use (AOD (Line 66), NRT (Line 157).

2.       The reader would benefit from a brief overview of the intervention in the introductory section.

3.       Inclusion of the interview/focus group discussion guides would be useful (this could be included in supplementary material, if needed).

4.      Please say how staff were recruited?  In addition, was there a standardised procedure for client recruitment?

5.       Please provide information on consent procedures e.g. was written consent obtained and separate consent for audio? 

6.      Given the small sample, the reporting of proportions is not required. I do not think there is a need for a response rate. Simply reporting that x consented out of x invited is appropriate. 

7.       In the tables reporting participant demographic and characteristics.  (1) the small frequencies do not require proportions to be reported; (2) the SD for mean participant age and employment histories are reported under the proportions column.  I would suggest reporting mean and SD in the text, instead of in the table. 

8.      In the discussion, I wondered if the findings could be compared to the literature from other settings, such as mental health or other disadvantaged or high-risk populations. Drawing a comparison would be both interesting and helpful.

9.      In the strengths and limitations sub-section, Line 345, please include that other facilities nationally or internationally may not have the same experience.   

10.    If word counts allow a short reflexive statement would be appreciated and would be of interest to other researchers contemplating similar work. 

Author Response

Sunday 2nd June 2019

Section Managing Editor

Dear Ms Chen

Thank you for your email dated  Wednesday 29th May 2019 seeking a revision of our manuscript ijerph-519375 “Integrating smoking cessation care into a medically supervised injecting facility using an organizational change intervention: A qualitative study of staff and client views for the journal International Journal of Environmental Research and Public Health. We thank the reviewers who have allowed us to improve the manuscript through their suggestions. Please find below responses to the requested changes. Additionally, the amendments can be viewed in the manuscript and are in track changes.

REVIEWER ONE

1.            Abstract. Background and aims – I would alter ‘alcohol and other drug (AOD)’ to healthcare

The sentence now reads: “Organisational change interventions such as the Addressing Tobacco Through Organisational Change [6, 7] and Systems Change Approach [8] have been developed for increasing smoking cessation care delivery in the alcohol and other drug (AOD) setting.”

2.            The reader would benefit from a brief overview of the intervention in the introductory section.

Introduction section, third paragraph now reads: “An organisational change intervention was implemented. The intervention consisted of six core components: engagement of organisational support, identification of a support champion, providing education and resources, provide evidence based treatments, creation of a system to assess and record smoking status and promotion of centre policies that support and provide tobacco dependence services.”

3.            Inclusion of the interview/focus group discussion guides would be useful (this could be included in supplementary material, if needed)

Files are provided as supplementary materials (1, 2,3). Reference to these files is made in section 2.3.

4.            Please say how staff were recruited? In addition, was there a standardised procedure for client recruitment?

The following information has been added to section 2.3 Procedures:

“The site contact disseminated an email invitation with a participant information statement attached to all members of staff (variety of management and staff roles, those who work weekdays and weekend for a representative sample). Individuals interested in participating were able to email or call to register interest in participating in the interviews."

"A flyer promoting the client focus group was displayed in the waiting room and after care rooms of the facility."

5.            Please provide information on consent procedures e.g. was written consent obtained and separate consent for audio?

The authors have added the following to section ‘2.3 Procedures’:

“Written consent was gained for the demographics survey and audio recording prior to study commencement.”

6.            In the strengths and limitations sub-section, Line 345, please include that other facilities nationally or internationally may not have the same experience.  

The sentence now reads:

“First, the participants were all staff and clients from a single Australian SIF thus their experiences may not occur for facilities located nationally or internationally.”

7.            If word counts allow a short reflexive statement would be appreciated and would be of interest to other researchers contemplating similar work.

The authors thank the reviewers for their suggestion and have chosen not to add the reflexive statement.

 We hope you find these modifications and explanations satisfactory. We look forward to your response and the opportunity to publish with the International Journal of Environmental Research and Public Health.

 Kind regards,

 Reviewer 2 Report

The subject of the manuscript is of great interest since the tobacco epidemic continues to be a serious public health problem and companies and institutions are not yet fully involved.

The language used is clear, the results are very understandable.

Only in my view is not enough emphasis on the limitations in relation to the size of the sample that although it is a study with qualitative techniques, the sample is very small.

Author Response

Sunday 2nd June 2019

Thank you for your email dated  Wednesday 29th May 2019 seeking a revision of our manuscript ijerph-519375 “Integrating smoking cessation care into a medically supervised injecting facility using an organizational change intervention: A qualitative study of staff and client views for the journal International Journal of Environmental Research and Public Health. We thank the reviewers who have allowed us to improve the manuscript through their suggestions. Please find below responses to the requested change.

REVIEWER TWO

1.      Only in my view is not enough emphasis on the limitations in relation to the size of the sample that although it is a study with qualitative techniques, the sample is very small.

 The authors thank the reviewer for their comments. As per best practice qualitative procedures, data collection continued until saturation was reached. This is alluded to in the abstract section of the manuscript

 We look forward to your response and the opportunity to publish with the International Journal of Environmental Research and Public Health.

 Kind regards,